# The association between alcohol consumption and herpes simplex virus type 2: A cross-sectional study from national health and nutrition examination survey 2009–2016

Yushan Shi[1,2], Jiafeng Zhang[3], Zhantong Wang[4], Feng Shan[1]*

1 Department of Orthopaedics, Children's Hospital of Soochow University, Suzhou, China, 2 Department of Laboratory, Affiliated Hospital of Shandong University of Traditional Chinese Medicine, Jinan, Shandong, China, 3 Department of Laboratory Medicine, Shanghai Changzheng Hospital, Naval Medical University, Shanghai, P.R. China, 4 Department of General Surgery, Naval Medical Center of PLA, Naval Medical University, Shanghai, China

☯ These authors contributed equally to this work.
* maple19920719@163.com

**Data Availability Statement:** All data are available from https://www.cdc.gov/nchs/nhanes/index.htm.

## Abstract

### Background

The current prevalence of Herpes simplex virus type 2 (HSV-2) infection is notably high, with individuals afflicted by HSV-2 facing recurrent outbreaks, challenges in achieving remission, and an elevated risk of HIV infection. This study aims to investigate the relationship between alcohol consumption and HSV-2 infection.

### Methods

The data for this study were sourced from 7257 participants who took part in the National Health and Nutrition Examination Survey (NHANES) from 2009 to 2016. The target population consisted of adults with reliable HSV-2 plasma results, and alcohol consumption was assessed using self-report methods. We evaluated the odds ratio (OR) and 95% confidence interval (CI) for the association between alcohol consumption and HSV-2 infection. These estimations were derived from a logistic regression model that was adjusted for key confounding factors. Subgroup analysis specifically focused on alcohol consumption, and the interaction between HSV-2 infection, alcohol consumption, and other variables was assessed through stratified analysis.

### Results

Among the 7,257 participants included, 89.8% (6,518/7,257) reported varying levels of alcohol consumption history. Compared to individuals who never drinkers, the adjusted odds ratios (ORs) for former drinkers, light drinkers, moderate drinkers, and heavy drinkers were 1.79 (95% CI: 1.34–2.4, p < 0.001), 1.38 (95% CI: 1.07–1.77, p = 0.012), 1.49 (95% CI: 1.15–1.94, p = 0.003), and 1.47 (95% CI: 1.14–1.9, p = 0.003), respectively. The results remained stable in subgroup analyses and sensitivity analyses.

**Funding:** This study was supported by National Natural Science Foundation of China (882172520), Soochow University High-End Platform and Translational Base Construction Project for Clinical Medicine Science and Technology.

**Competing interests:** The authors have declared that no competing interests exist.

**Abbreviations:** AChR, α-acetylcholine receptors; BMI, Body Mass Index; CI, Confidence interval; EBV, Epstein-Barr virus; HbA1c, Glycated hemoglobin; HIV, Human immunodeficiency virus; HSV, Herpes Simplex Virus; NCHS, National Center for Health Statistics; NHANES, National Health and Nutrition Examination Survey; OR, Odds ratio; PSM, Propensity score matching; SD, Standard deviation; Std diff, Standardized difference.

## Conclusion

Current research indicates that individuals with a history of alcohol consumption exhibit a higher risk of HSV-2 infection compared to those who have never drinkers.

## 1. Introduction

Herpes Simplex Virus type 2 (HSV-2) is the primary causative agent of genital herpes, a ulcerative condition characterized by recurrent, lingering outbreaks and an increased risk of HIV infection [1]. The prevalence of HSV-2 varies across different countries and regions, exhibiting a rising trend. The World Health Organization (WHO) estimates that over 500 million individuals aged 15–49 globally are infected with HSV-2, with an annual increment of 24 million new cases [2, 3]. HSV-2 is primarily transmitted through sexual contact, via exposure to infected genitalia or bodily fluids, underscoring the impact of lifestyle factors on this mode of transmission. Hence, investigating the influence of lifestyle factors related to HSV infection is crucial [4].

Alcohol consumption, a common behavior in various lifestyles, has been shown to significantly affect life expectancy and contribute to the global disease and mortality burden, potentially becoming the third-largest modifiable risk factor for death and disability worldwide [5]. While moderate drinking is considered beneficial for certain diseases, other studies report no benefits [6]. The relationship between alcohol consumption and HSV-2 infection in a representative American population remains unclear.

Using data from the National Health and Nutrition Examination Survey (NHANES), this study evaluates the relationship between different alcohol consumption statuses and HSV-2 infection, aiming to fill this knowledge gap. We detailed alcohol intake levels and analyzed the association between different drinking statuses and HSV-2. Additionally, we explored whether these associations varied by age, gender, ethnicity, and other covariates. The insights gathered could enhance the broader understanding of the impact of lifestyle factors, particularly alcohol consumption, on HSV-2 transmission.

## 2. Methods

### 2.1 Study design and participants

This cross-sectional study utilized data from 4 consecutive cycles of the NHANES spanning from 2009 to 2016. The NHANES was authorized by the Ethical Review Committee of the National Center for Health Statistics, and all participants provided written informed consent. The research reported here employed publicly available and de-identified data; thus, it was exempt from review and the requirement of informed consent. This study adhered to the STROBE (Strengthening the Reporting of Observational studies in Epidemiology) reporting guidelines [7].

The NHANES is a cross-sectional survey that utilizes stratified, multi-stage random sampling techniques to collect data from a nationally representative sample of non-institutionalized citizens of the United States. Among 40439 participants, we excluded individuals for 1) missing HSV-2 data (n = 28612), 2) absence of alcohol consumption data (n = 1528), and 3) missing data for potential covariates, including marital status (n = 767), ratio of family income to poverty (PIR) (n = 692), education level (n = 5), body mass index (BMI) (n = 44), health insurance coverage (n = 7), smoking status (n = 5), frequency of condom use (n = 1,312),

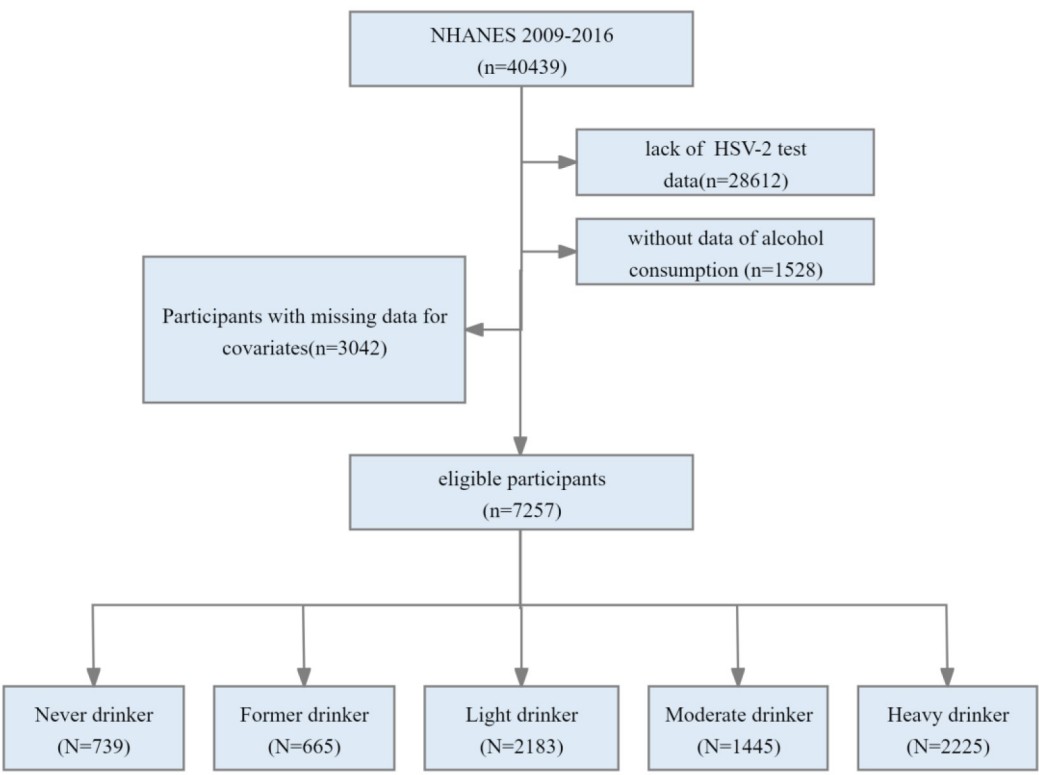

**Fig 1. The study's flow diagram.** Abbreviations: HSV, herpes simplex virus; NHANES, National Health and Nutrition Examination Survey.

diabetes (n = 180), hypertension (n = 1), chronic kidney disease (CKD) (n = 28), and chronic obstructive pulmonary disease (COPD) (n = 1). (Fig 1).

## 2.2 Alcohol consumption

Trained interviewers used the Computer-Assisted Personal Interviewing (CAPI) system to inquire about alcohol intake. Based on the self-reported responses from these questionnaires, participants were categorized into 5 alcohol consumption as described in previous studies [8, 9].

Participants were categorized into distinct alcohol consumption groups based on their drinking patterns: 1) never drinkers (consumed less than 12 alcoholic beverages in their lifetime); 2) Former drinkers (consumed 12 or more alcoholic beverages in a year but abstained last year, or abstained last year but had a lifetime consumption of 12 or more drinks); 3) light drinkers (up to 1 drink per day for women and up to 2 drinks per day for men on average over the past year); 4) moderate drinkers (up to 2 drinks daily for women and up to 3 for men); 5) heavy drinkers (3 or more drinks daily for women, 4 or more for men).

## 2.3 Definition of HSV-2

Participants provided serum samples at the time of the survey, which were stored under appropriate frozen conditions (-30°C) and sent to a Clinical Laboratory Improvement Amendments (CLIA) certified laboratory. A complete summary of the laboratory methods is available on the NHANES website.

Through the use of monoclonal antibodies and affinity chromatography, a specific glycoprotein known as gG-2 is purified from HSV-2. The solid-phase enzymatic immune dot assay is then employed to detect antibodies reactive to the gG-2 antigen in the serum.

### 2.4 Study covariates

In this study, the covariates considered included age, gender, self-reported race and ethnicity (categories being Mexican American, non-Hispanic Black, non-Hispanic White, and others), marital status (either living alone or married/living with a partner), educational attainment (categorized as less than high school, high school or GED equivalent, and above high school), PIR ($\leq 1.3$, 1.3–3.5, and >3.5), BMI (<25 and $\geq 25$), health insurance coverage (yes or no), smoking status (never, former, or current smoker), frequency of condom use (never, less than half the time, about half the time, more than half the time, always). The frequency of condom use was determined based on responses to variable SXQ251.

Diabetes was defined as a fasting blood glucose level $\geq 7$ mmol/L, a medical diagnosis of diabetes by a physician, use of oral hypoglycemic agents or insulin, or a glycated hemoglobin (HbA1c) level $\geq 6.5\%$ [10]. Hypertension was defined as having a systolic blood pressure of >130 mmHg or a diastolic blood pressure of >80 mmHg, averaged over three measurements, or a history of high blood pressure or oral antihypertensive medication use [11]. CKD was defined based on an eGFR lower than 60 ml/min/1.73m^2 or a urine albumin-to-creatinine ratio of 30 mg/g or higher, in line with the KDIGO 2021 guidelines for Glomerular Diseases [12]. COPD criteria included: 1) A FEV1/FVC ratio below 0.7; 2) A previous diagnosis of emphysema; 3) Being over 40 years old, with a smoking history or chronic bronchitis, and the use of specific medications such as phosphodiesterase-4 inhibitors [13]. Cardiovascular Disease (CVD) and cancer were considered present if participants were informed by a physician of having such conditions.

### 2.5 Statistical analyses

Data analysis was conducted from October 1 to December 31, 2023. Considering the complex sampling methods of NHANES. Baseline characteristics for each alcohol consumption category were compared using the chi-square test for categorical variables and analysis of variance for continuous variables. Logistic regression models were employed to determine the odds ratios (OR) and 95% confidence intervals (95% CI) for the association between alcohol consumption and HSV-2. Model 1 was adjusted for age, gender, race/ethnicity, marital status, and education level; Model 2 additionally adjusted for PIR, smoking status, frequency of condom use, while Model 3 included adjustments for all the variables incorporated.

In the subgroup analyses, we replicated the primary analyses, stratifying alcohol consumption by sex, age (<31 years vs $\geq 31$ years), marital status, PIR (low vs medium or high), diabetes (yes vs no), hypertension (yes vs no).

In the sensitivity analyses, we categorized alcohol consumption into never drinkers and drinkers, and conducted sensitivity analyses across the previously described 3 models.

Statistical analyses were conducted using R software (version 4.2.2, R Foundation for Statistical Computing) and Free Statistics software version 1.8. All hypothesis tests were two-sided, with a *P*-value of less than 0.05 considered to indicate statistical significance.

## 3. Results

### 3.1 Study population characteristics

The analytic sample comprised 7257 participants (mean [SD] age 34.5 [8.5] years; 3643 women [50.2%] and 3614 men [49.8%]; 1131 Mexican Americans [15.6%], 1414 Black

[19.5%], 3033 White [41.8%], and 1679 of other races and ethnicities [23.1%]). The positivity individuals for HSV-2 was 1464 (20.2%). Detailed participant demographics stratified by alcohol consumption are presented in Table 1. 10.2% (739) of participants were never drinkers, 9.2% (665) were former drinkers, 30.1% (2183) were light drinkers, 19.9% (1445) were moderate drinkers, and 30.7% (2225) were heavy drinkers. Statistical significant differences were observed in baseline parameters across the five alcohol consumption groups. Compared to other groups, former drinkers had the highest proportion of HSV-2 positivity (26.2%). Relative to never drinkers, heavy drinkers had a higher proportion of males, a higher frequency of smoking, and higher rates of hypertension, CKD, COPD, CVD, and cancer.

Univariate analysis of all variables in Table 1 indicated that age, gender, marital status, PIR, BMI, health insurance, smoking status, and comorbidities are associated with HSV infection. (S1 Table in S1 File).

### 3.2 Relationship between alcohol consumption and HSV-2

In the expanded multivariate logistic regression models, we observed that the odds ratios (OR) for alcohol consumption were consistently significant across all three models (OR range 1.36–2.14, p < 0.05). Without adjusting any factors in the crude model, former drinkers and moderate drinkers exhibited a higher risk of HSV-2 infection (OR = 1.69, 95% CI: 1.31–2.19, p < 0.001; OR = 1.37, 95% CI: 1.09–1.72, p = 0.007). After adjustments for age, sex, race/ethnicity, marital status, and education level, alcohol consumption in all four statuses remained associated with an increased risk (OR range 1.42–2.14, p < 0.05). The results of the fully adjusted model 3 were consistent with earlier findings (OR range 1.38–1.79, p < 0.05). Across all three models, the highest risk of HSV-2 positivity was consistently observed in the former drinkers group compared to the never drinkers group, with OR ranging from 1.69 to 2.14 (p<0.001). (Table 2).

In the subgroup analyses stratified by age group, gender, marital status, poverty income ratio, diabetes, and hypertension, the association between alcohol consumption and HSV-2 infection was consistent (Fig 2). This consistent association was also observed in other subgroups (S1 Fig in S1 File).

In the sensitivity analyses, after categorizing alcohol consumption into two groups: never drinkers and drinkers, it was found that in the crude model, without adjusting for any variables, the risk of HSV-2 infection was higher in the drinking group (OR = 1.23, 95% CI: 1.01–1.5, p = 0.042). Across all three models, compared to never drinkers, the drinking group consistently showed a higher risk of HSV-2 infection (Table 3).

## 4. Discussion

This cross-sectional study demonstrates an association between alcohol consumption and the risk of HSV-2 infection. In our multivariate logistic regression models, both former and current drinkers exhibited a higher risk of HSV-2 infection compared to individuals who have never consumed alcohol. Our findings consistently indicate that alcohol consumption, whether past or present, is a risk factor for HSV-2 infection.

The impact of alcohol consumption on HSV-2 infection has been documented in limited research. A study by Trong et al. suggests that reducing alcohol intake could effectively control HSV-2 infection among bar and hotel workers [14]. Similarly, research by Yasufumi et al. found that current male drinkers in Japan are more susceptible to HSV-2 infection compared to never drinkers [15]. Notably, these studies, conducted outside the United States, do not extensively investigate the influence of sexual behavior on the relationship between alcohol consumption and HSV-2 infection. Utilizing NHANES data, our study offers a unique

Table 1. Baseline demographic characteristics of the participants according to alcohol consumption.

| Variables | Total (n = 7257) | Never drinkers (n = 739) | Former drinkers (n = 665) | Light drinkers (n = 2183) | Moderate drinkers (n = 1445) | Heavy drinkers (n = 2225) | p |
|---|---|---|---|---|---|---|---|
| **Age(years)** | 34.5 ± 8.5 | 35.0 ± 8.6 | 37.3 ± 8.1 | 35.5 ± 8.3 | 34.5 ± 8.5 | 32.7 ± 8.6 | < 0.001 |
| **Age(years), n (%)** | | | | | | | < 0.001 |
| 18–24 | 1173 (16.2) | 112 (15.2) | 61 (9.2) | 271 (12.4) | 221 (15.3) | 508 (22.8) | |
| 25–30 | 1444 (19.9) | 138 (18.7) | 84 (12.6) | 388 (17.8) | 309 (21.4) | 525 (23.6) | |
| 31–40 | 2469 (34.0) | 258 (34.9) | 251 (37.7) | 798 (36.6) | 485 (33.6) | 677 (30.4) | |
| 41–49 | 2171 (29.9) | 231 (31.3) | 269 (40.5) | 726 (33.3) | 430 (29.8) | 515 (23.1) | |
| **Gender, n (%)** | | | | | | | < 0.001 |
| Female | 3643 (50.2) | 504 (68.2) | 339 (51) | 936 (42.9) | 909 (62.9) | 955 (42.9) | |
| Male | 3614 (49.8) | 235 (31.8) | 326 (49) | 1247 (57.1) | 536 (37.1) | 1270 (57.1) | |
| **Race/Ethnicity, n (%)** | | | | | | | < 0.001 |
| Mexican American | 1131 (15.6) | 143 (19.4) | 110 (16.5) | 222 (10.2) | 194 (13.4) | 462 (20.8) | |
| Non-Hispanic black | 1414 (19.5) | 175 (23.7) | 97 (14.6) | 496 (22.7) | 309 (21.4) | 337 (15.1) | |
| Non-Hispanic white | 3033 (41.8) | 191 (25.8) | 302 (45.4) | 899 (41.2) | 652 (45.1) | 989 (44.4) | |
| Others | 1679 (23.1) | 230 (31.1) | 156 (23.5) | 566 (25.9) | 290 (20.1) | 437 (19.6) | |
| **Marital status, n (%)** | | | | | | | < 0.001 |
| Living alone | 2592 (35.7) | 177 (24) | 149 (22.4) | 654 (30) | 558 (38.6) | 1054 (47.4) | |
| Married or living with a partner | 4665 (64.3) | 562 (76) | 516 (77.6) | 1529 (70) | 887 (61.4) | 1171 (52.6) | |
| **Education level, n (%)** | | | | | | | < 0.001 |
| Less than high school | 1236 (17.0) | 160 (21.7) | 159 (23.9) | 233 (10.7) | 186 (12.9) | 498 (22.4) | |
| High school or GED | 1564 (21.6) | 157 (21.2) | 173 (26) | 368 (16.9) | 281 (19.4) | 585 (26.3) | |
| Above high school | 4457 (61.4) | 422 (57.1) | 333 (50.1) | 1582 (72.5) | 978 (67.7) | 1142 (51.3) | |
| **PIR, n (%)** | | | | | | | < 0.001 |
| ≤1.3 | 2469 (34.0) | 323 (43.7) | 295 (44.4) | 577 (26.4) | 412 (28.5) | 862 (38.7) | |
| 1.3–3.5 | 2603 (35.9) | 249 (33.7) | 236 (35.5) | 759 (34.8) | 532 (36.8) | 827 (37.2) | |
| >3.5 | 2185 (30.1) | 167 (22.6) | 134 (20.2) | 847 (38.8) | 501 (34.7) | 536 (24.1) | |
| **BMI(Kg/m2)** | | | | | | | < 0.001 |
| <25 | 2378 (32.8) | 228 (30.9) | 170 (25.6) | 784 (35.9) | 500 (34.6) | 696 (31.3) | |
| ≥25 | 4879 (67.2) | 511 (69.1) | 495 (74.4) | 1399 (64.1) | 945 (65.4) | 1529 (68.7) | |
| **Health Insurance Coverage, n (%)** | | | | | | | < 0.001 |
| No | 2157 (29.7) | 236 (31.9) | 201 (30.2) | 491 (22.5) | 376 (26) | 853 (38.3) | |
| Yes | 5100 (70.3) | 503 (68.1) | 464 (69.8) | 1692 (77.5) | 1069 (74) | 1372 (61.7) | |
| **Smoking status, n (%)** | | | | | | | < 0.001 |
| Never | 4350 (59.9) | 658 (89) | 369 (55.5) | 1497 (68.6) | 844 (58.4) | 982 (44.1) | |
| Former | 1104 (15.2) | 25 (3.4) | 133 (20) | 334 (15.3) | 259 (17.9) | 353 (15.9) | |
| Current | 1803 (24.8) | 56 (7.6) | 163 (24.5) | 352 (16.1) | 342 (23.7) | 890 (40) | |
| **Frequency of Condom Use, n (%)** | | | | | | | < 0.001 |
| Never | 1737 (23.9) | 229 (31) | 165 (24.8) | 499 (22.9) | 328 (22.7) | 516 (23.2) | |
| Less than half of time | 1052 (14.5) | 107 (14.5) | 70 (10.5) | 313 (14.3) | 198 (13.7) | 364 (16.4) | |
| About half of time | 495 (6.8) | 35 (4.7) | 40 (6) | 134 (6.1) | 89 (6.2) | 197 (8.9) | |
| More than half of time | 681 (9.4) | 56 (7.6) | 57 (8.6) | 172 (7.9) | 165 (11.4) | 231 (10.4) | |
| Always | 3292 (45.4) | 312 (42.2) | 333 (50.1) | 1065 (48.8) | 665 (46) | 917 (41.2) | |
| **Diabetes, n (%)** | | | | | | | < 0.001 |
| No | 6908 (95.2) | 696 (94.2) | 610 (91.7) | 2078 (95.2) | 1386 (95.9) | 2138 (96.1) | |
| Yes | 349 (4.8) | 43 (5.8) | 55 (8.3) | 105 (4.8) | 59 (4.1) | 87 (3.9) | |
| **Hypertension, n (%)** | | | | | | | < 0.001 |
| No | 5696 (78.5) | 601 (81.3) | 483 (72.6) | 1725 (79) | 1154 (79.9) | 1733 (77.9) | |

*(Continued)*

**Table 1.** (Continued)

| Variables | Total (n = 7257) | Never drinkers (n = 739) | Former drinkers (n = 665) | Light drinkers (n = 2183) | Moderate drinkers (n = 1445) | Heavy drinkers (n = 2225) | p |
|---|---|---|---|---|---|---|---|
| Yes | 1561 (21.5) | 138 (18.7) | 182 (27.4) | 458 (21) | 291 (20.1) | 492 (22.1) | |
| **CKD, n (%)** | | | | | | | 0.256 |
| No | 6769 (93.3) | 689 (93.2) | 609 (91.6) | 2053 (94) | 1348 (93.3) | 2070 (93) | |
| Yes | 488 (6.7) | 50 (6.8) | 56 (8.4) | 130 (6) | 97 (6.7) | 155 (7) | |
| **COPD, n (%)** | | | | | | | 0.118 |
| No | 7138 (98.4) | 733 (99.2) | 651 (97.9) | 2150 (98.5) | 1425 (98.6) | 2179 (97.9) | |
| Yes | 119 (1.6) | 6 (0.8) | 14 (2.1) | 33 (1.5) | 20 (1.4) | 46 (2.1) | |
| **CVD, n (%)** | | | | | | | < 0.001 |
| No | 7073 (97.5) | 728 (98.5) | 633 (95.2) | 2134 (97.8) | 1415 (97.9) | 2163 (97.2) | |
| Yes | 184 (2.5) | 11 (1.5) | 32 (4.8) | 49 (2.2) | 30 (2.1) | 62 (2.8) | |
| **Cancer, n (%)** | | | | | | | 0.331 |
| No | 7065 (97.4) | 725 (98.1) | 647 (97.3) | 2130 (97.6) | 1397 (96.7) | 2166 (97.3) | |
| Yes | 192 (2.6) | 14 (1.9) | 18 (2.7) | 53 (2.4) | 48 (3.3) | 59 (2.7) | |
| **HSV-2, n (%)** | | | | | | | < 0.001 |
| Negative | 5793 (79.8) | 611 (82.7) | 491 (73.8) | 1776 (81.4) | 1123 (77.7) | 1792 (80.5) | |
| Positive | 1464 (20.2) | 128 (17.3) | 174 (26.2) | 407 (18.6) | 322 (22.3) | 433 (19.5) | |

Abbreviations: PIR Ratio of family income to poverty, BMI Body mass index, CKD Chronic kidney disease, CVD Cardiovascular disease, COPD Chronic obstructive pulmonary disease, HSV herpes simplex virus; IQR Interquartile Range.

opportunity to assess this association with comprehensive adjustments for a wide range of covariates, including sexual activity and frequency of condom use, through a series of stratified analyses.

While the underlying mechanisms of the association between alcohol consumption and HSV-2 remain to be further explored, our findings are biologically plausible based on existing evidence. Previous studies have shown that drinking can increase high-risk behaviors, thereby raising the risk of sexually transmitted diseases (STDs), whereas limiting or abstaining from alcohol can reduce such behaviors and lower the disease burden of STDs [16, 17]. Alcohol affects the neurotransmitter and neuroendocrine systems, leading to disruptions in emotional perception and cognitive functions, which could be the reason for the increase in risky behaviors following alcohol consumption [18, 19]. This, in turn, elevates the risk of HSV-2 infection.

**Table 2.** Association between alcohol consumption and HSV-2.

| Alcohol consumption | Crude model | | Model 1 | | Model 2 | | Model 3 | |
|---|---|---|---|---|---|---|---|---|
| | OR (95% CI) | p | OR (95% CI) | p | OR (95% CI) | p | OR (95% CI) | p |
| Never drinkers | Ref | | Ref | | Ref | | Ref | |
| Former drinkers | 1.69 (1.31~2.19) | <0.001 | 2.14 (1.61~2.85) | <0.001 | 1.82 (1.36~2.44) | <0.001 | 1.79 (1.34~2.4) | <0.001 |
| Light drinkers | 1.09 (0.88~1.36) | 0.421 | 1.42 (1.12~1.82) | 0.005 | 1.36 (1.06~1.74) | 0.016 | 1.38 (1.07~1.77) | 0.012 |
| Moderate drinkers | 1.37 (1.09~1.72) | 0.007 | 1.65 (1.28~2.13) | <0.001 | 1.49 (1.15~1.93) | 0.003 | 1.49 (1.15~1.94) | 0.003 |
| Heavy drinkers | 1.15 (0.93~1.43) | 0.198 | 1.84 (1.44~2.35) | <0.001 | 1.49 (1.16~1.93) | 0.002 | 1.47 (1.14~1.9) | 0.003 |

Crude model adjusted for: none

Model 1 adjusted for: age, sex, race/ethnicity, marital status, education level

Model 2 adjusted for: age, sex, race/ethnicity, marital status, education level, PIR, smoking status, frequency of condom use

Model 3 adjusted for: all variables

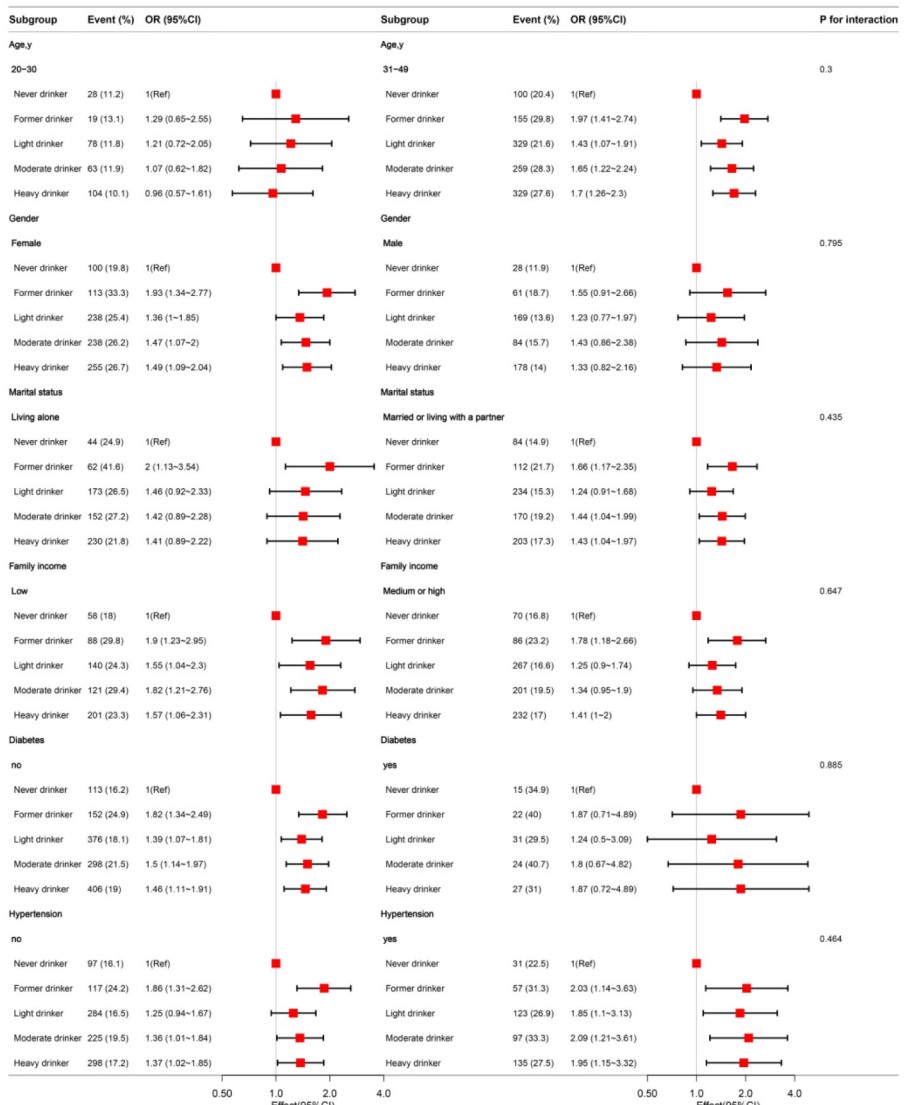

**Fig 2. The relationship between alcohol consumption and HSV-2 according to basic features.** Except for the stratification component itself, each stratification factor was adjusted for all other variable.

**Table 3. Sensitivity analysis of the relationship between alcohol consumption and HSV-2 infection.**

| Alcohol consumption | Crude model | | Model 1 | | Model 2 | | Model 3 | |
|---|---|---|---|---|---|---|---|---|
| | OR (95% CI) | p | OR (95% CI) | p | OR (95% CI) | p | OR (95% CI) | p |
| Never drinkers | Ref | | Ref | | Ref | | Ref | |
| drinkers | 1.23 (1.01~1.5) | 0.042 | 1.68 (1.34~2.1) | <0.001 | 1.48 (1.18~1.87) | 0.001 | 1.48 (1.18~1.86) | <0.001 |

Crude model adjusted for: none

Model 1 adjusted for: age, sex, race/ethnicity, marital status, education level

Model 2 adjusted for: age, sex, race/ethnicity, marital status, education level, PIR, smoking status, frequency of condom use

Model 3 adjusted for: all variables

Furthermore, alcohol use, regardless of the amount, leads to health detriment in populations, and long-term drinking alters adaptive immunity and cytokine activity, affecting inflammatory responses [20, 21]. Moreover, studies have shown that chronic alcohol consumption can alter the transcriptome of bone marrow-derived monocytes, potentially impacting monocyte activity and their migration from the bone marrow, thereby affecting immune responses. Additionally, chronic alcohol intake can bias the differentiation of CD34+ progenitor cells in the bone marrow towards the granulocyte/monocyte lineage, leading to a decreased antigen-presenting capability of monocytes derived from CD34+ progenitor cells [22]. This alteration persists even after cessation of alcohol consumption and may compromise both immune and inflammatory responses. These impacts on the immune system and inflammatory responses could increase susceptibility to HSV-2, also explaining why former drinkers may still exhibit a heightened risk of HSV-2 infection.

Our study is subject to several limitations that warrant consideration, which may affect the interpretation and generalizability of the findings. Firstly, although we observed the highest risk of HSV-2 infection among former drinkers, this finding is based on the assumption that most of them were heavy drinkers before they quit. However, the NHANES dataset lacks specific data regarding the average daily alcohol consumption of former drinkers prior to cessation. This limitation restricts our ability to fully understand their previous drinking habits and its potential impact on the risk of HSV-2 infection. Secondly, despite utilizing regression models and stratified analyses to mitigate the effects of confounding variables, the potential for residual confounding from unmeasured or unknown factors cannot be entirely eliminated. This limitation underscores the need for cautious interpretation of the associations observed, as other unaccounted variables might influence the outcomes. Thirdly, our findings, based on a survey of U.S. adults, may not be generalizable to other populations without further investigation. Cultural, social, and behavioral differences in various populations might affect the relationship between alcohol consumption and HSV-2, thus limiting the universality of our results. Lastly, the inherent limitations of cross-sectional studies constrain our ability to establish a causal relationship between alcohol consumption and HSV-2 infection. Longitudinal studies are required to confirm these findings and to explore the temporal sequence between alcohol consumption and HSV-2 infection. Moreover, although our study focuses on the association between alcohol consumption and HSV-2, other lifestyle factors such as smoking also play a significant role in HSV-2 infection and warrant further exploration to fully understand the multifactorial nature of this health issue.

## 5. Conclusion

In conclusion, there is an association between alcohol consumption and HSV-2 infection among adult populations in the United States. The results of this study have drawn attention to the link between lifestyle factors, particularly alcohol consumption, and HSV-2 infection.

## Supporting information

**S1 File.**
(DOCX)

## Acknowledgments

Thanks to Zhang Jing (Second Department of Infectious Disease, Shanghai Fifth People's Hospital, Fudan University) for his work on the NHANES database.

## Author Contributions

**Conceptualization:** Jiafeng Zhang, Feng Shan.

**Data curation:** Yushan Shi, Jiafeng Zhang, Zhantong Wang, Feng Shan.

**Formal analysis:** Jiafeng Zhang.

**Funding acquisition:** Feng Shan.

**Investigation:** Yushan Shi, Feng Shan.

**Methodology:** Feng Shan.

**Project administration:** Yushan Shi.

**Resources:** Zhantong Wang, Feng Shan.

**Software:** Zhantong Wang, Feng Shan.

**Supervision:** Feng Shan.

**Validation:** Feng Shan.

**Writing – original draft:** Yushan Shi, Jiafeng Zhang, Zhantong Wang, Feng Shan.

**Writing – review & editing:** Feng Shan.

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
