## [Decision Letter · Decision Letter 0]

24 May 2024

PONE-D-24-11862

The association between alcohol consumption and Herpes simplex virus type 2: A cross-sectional study from National Health and Nutrition Examination Survey 2009-2016

PLOS ONE

Dear Dr. Shan,

Thank you for submitting your manuscript to PLOS ONE. After careful consideration, we feel that it has merit but does not fully meet PLOS ONE’s publication criteria as it currently stands. Therefore, we invite you to submit a revised version of the manuscript that addresses the points raised during the review process.

We look forward to receiving your revised manuscript.

Kind regards,

Tinashe Mudzviti, MPhil(MD)

Academic Editor

PLOS ONE

Journal Requirements:

   "no"

6. Please include your tables as part of your main manuscript and remove the individual files. Please note that supplementary tables (should remain/ be uploaded) as separate "supporting information" files

Additional Editor Comments (if provided):

Reviewers' comments:

Reviewer's Responses to Questions

**Comments to the Author**

1. Is the manuscript technically sound, and do the data support the conclusions?

Reviewer #1: Yes

Reviewer #2: Yes

2. Has the statistical analysis been performed appropriately and rigorously? 

Reviewer #1: Yes

Reviewer #2: Yes

3. Have the authors made all data underlying the findings in their manuscript fully available?

Reviewer #1: Yes

Reviewer #2: Yes

4. Is the manuscript presented in an intelligible fashion and written in standard English?

Reviewer #1: Yes

Reviewer #2: Yes

5. Review Comments to the Author

Reviewer #1: Thank you for taking the time to conduct this cross-sectional study. In the methods section, you categorized and defined alcohol consumption into five groups: non-drinkers, former drinkers, current light drinkers, current moderate drinkers, and heavy drinkers. Regarding the non-drinkers group, throughout the study, you use non-drinkers interchangeably with never-drinkers. However, the non-drinkers group is defined as those who “consumed less than 12 alcoholic beverages in their lifetime.” Should this be updated for consistency and accuracy across the study?

Reviewer #2: I would suggest the following minor revisions to improve the clarity, precision, and overall quality of the manuscript:

Clarification of Terminology: Ensure that all terms, especially those related to alcohol consumption categories, are clearly defined and consistent throughout the manuscript.

Expansion on Limitations: While the authors have mentioned the limitations, it would be beneficial to expand on how these limitations might impact the interpretation of the results and what that means for the generalizability of the findings.

Additional Subgroup Analysis: Consider including additional subgroup analyses based on other potential confounders such as socioeconomic status or geographic location to further explore the heterogeneity of the results.

Strengthening the Discussion: The discussion section could be strengthened by incorporating a more detailed comparison with existing literature, especially regarding the biological mechanisms by which alcohol consumption might increase the risk of HSV-2 infection.

Statistical Methodology: Provide more detail on the statistical methods used, particularly regarding the adjustments made in the logistic regression models and the rationale behind the choice of these adjustments.

Consistency in Data Presentation: Ensure that tables and figures are consistent with the text and that they are clearly and accurately referenced within the manuscript.

Reference Updates: Check the references for the most current and relevant literature to support the study's findings and arguments.

Language and Grammar: While the manuscript is generally well-written, there may be minor grammatical errors or awkward phrasings that could be polished for better readability.

Confirmation of Ethical Compliance: If applicable, provide a statement confirming that the study complies with ethical standards regarding the protection of personal data and privacy of participants.

These minor revisions would help to enhance the manuscript's contribution to the field and ensure that it meets the high standards of the journal.

6. PLOS authors have the option to publish the peer review history of their article (what does this mean?). If published, this will include your full peer review and any attached files.

Reviewer #1: No

Reviewer #2: No

---

## [Author Response · Author response to Decision Letter 0]

21 Jun 2024

Response Letter

Dear Editor Tinashe Mudzviti,

On behalf of my co-authors, I greatly appreciate the careful review and insightful comments from the editor and the reviewers. We believe that by implementing the suggested changes, we now have a stronger manuscript entitled "The Association Between Alcohol Consumption and Herpes Simplex Virus Type 2: A Cross-Sectional Study from the National Health and Nutrition Examination Survey 2009-2016" (ID: PONE-D-24-11862) ready for submission to PLOS ONE. We look forward to your positive response to the revised work submitted here.

We present here point-to-point responses for each of the comments in the attached document and have revised our manuscript accordingly. We hope the revised manuscript will be acceptable. 

Editors' and reviewers' comments are given in bold, and specific points of advice have been numbered. Revised sections are identified with blue text in the manuscript.

There are no conflicts of interest regarding this work. All authors have read the revised manuscript and approved its submission to PLOS ONE. Please do not hesitate to contact us if further assistance is needed. 

Thank you and best regards.

Yours Sincerely,

Yushan Shi

Manuscript ID number: 

PONE-D-24-11862

Title of paper: 

The Association Between Alcohol Consumption and Herpes Simplex Virus Type 2: A Cross-Sectional Study from the National Health and Nutrition Examination Survey 2009-2016

Reply to Editor:

We have uploaded the three required documents: 'Response to Reviewers,' 'Revised Manuscript with Track Changes,' and the 'Manuscript' itself, in accordance with your guidelines. There is no need to amend the financial disclosure. We have resubmitted the figure files following the journal’s specifications. Our study does not require the submission of laboratory protocols.

Point-to-Point Responses

1.Please ensure that your manuscript meets PLOS ONE's style requirements, including those for file naming.

Response: 

We have revised our manuscript according to the PLOS ONE style templates provided on your website. This ensures that our submission meets all PLOS ONE style requirements, including those pertaining to file naming.

2.Please note that funding information should not appear in any section or other areas of your manuscript.

Response: 

Thank you for your reminder. We have removed the related information from the manuscript as per your request.

3.We note that the grant information you provided in the "Funding Information" and "Financial Disclosure" sections do not match. 

Response:

Thank you for your reminder. In the resubmitted manuscript, we have provided the accurate and final funding information.

4.Please complete your Competing Interests on the online submission form to state any Competing Interests.

Response:

We have indicated in our manuscript that "The authors have declared that no competing interests exist." We appreciate and agree to your updating the online submission form on our behalf.

5.Please ensure that you have an ORCID iD and that it is validated in Editorial Manager. 

Response:

Thank you for your guidance. We completed the process by updating our information and authenticating the ORCID iD as instructed.

6.Please include your tables as part of your main manuscript and remove the individual files.

Response:

We have removed the individual table files and included them as part of the main manuscript. Supplementary tables have been uploaded as separate "supporting information" files, as instructed.

7.Please include captions for your Supporting Information files at the end of your manuscript, and update any in-text citations to match accordingly.

Response:

We have added captions for the Supporting Information files at the end of our manuscript and updated the in-text citations accordingly.

8.Please review your reference list to ensure that it is complete and correct. 

Response:

We have reviewed our reference list and confirm that it does not include any retracted papers. Based on the reviewers' suggestions, we have added several relevant references. These additions have been highlighted in blue text in the 'Revised Manuscript with Track Changes.'

Request: Due to the recent job transition of the first author, we regretfully request to add another new affiliation in the author information. Please refer to line 7 for specific details.

Reply to Reviewer 1:

Dear Reviewer,

Thank you very much for taking the time to review our manuscript and for your constructive comments and valuable suggestions, which have greatly assisted in enhancing the quality of our work. We have carefully revised the manuscript in accordance with your feedback, thoroughly proofread the text, rephrased paragraphs, and corrected any inaccuracies.

We hope that these revisions will meet your approval, and we look forward to your acceptance of the amended manuscript.

1.In the methods section, you categorized and defined alcohol consumption into five groups: non-drinkers, former drinkers, current light drinkers, current moderate drinkers, and heavy drinkers. Regarding the non-drinkers group, throughout the study, you use non-drinkers interchangeably with never drinker. However, the non-drinkers group is defined as those who "consumed less than 12 alcoholic beverages in their lifetime." Should this be updated for consistency and accuracy across the study?

Response:

Thank you for your insightful comment regarding the classification of non-drinkers in our study. We have carefully reviewed the terminology used throughout the manuscript and have now standardized the term to "never drinkers" consistently across the text. This change clarifies our definition and aligns with your suggestion for accuracy and consistency. We appreciate your attention to detail and guidance, which have significantly improved the manuscript.

Reply to Reviewer 2:

Dear Reviewer,

Thank you very much for taking the time to review our manuscript. We are particularly grateful for your suggestions regarding Clarification of Terminology, Expansion on Limitations, Additional Subgroup Analysis, Strengthening the Discussion, Statistical Methodology, Consistency in Data Presentation, Reference Updates, Language and Grammar, and Confirmation of Ethical Compliance. Following your advice, we have thoroughly revised the manuscript. This includes standardizing terminology, thoroughly discussing limitations, adding subgroup analyses, and addressing the other issues you raised. 

We hope that the revised manuscript meets your approval, and we look forward to your acceptance.

1.Clarification of Terminology: Ensure that all terms, especially those related to alcohol consumption categories, are clearly defined and consistent throughout the manuscript.

Response:

Thank you for your constructive feedback. We have categorized and defined alcohol consumption into five groups: never drinkers, former drinkers, light drinkers, moderate drinkers, and heavy drinkers. We have carefully reviewed the terminology used throughout the manuscript to ensure that these five terms are standardized and consistently applied. This revision clarifies our definitions and aligns with your recommendations for accuracy and consistency. We appreciate your attention to detail and guidance, which have significantly enhanced the manuscript.

2.Expansion on Limitations: While the authors have mentioned the limitations, it would be beneficial to expand on how these limitations might impact the interpretation of the results and what that means for the generalizability of the findings.

Response:

Thank you for your suggestion. In response to your request, we have revised and, in fact, essentially rewritten the Limitations section of our manuscript. The new Limitations section has been expanded to include a detailed analysis of how each limitation might affect the interpretation of our results and their generalizability. The revised content is highlighted in blue in the manuscript.

Line 265-284：Our study is subject to several limitations that warrant consideration, which may affect the interpretation and generalizability of the findings. Firstly, although we observed the highest risk of HSV-2 infection among former drinkers, this finding is based on the assumption that most of them were heavy drinkers before they quit. However, the NHANES dataset lacks specific data regarding the average daily alcohol consumption of former drinkers prior to cessation. This limitation restricts our ability to fully understand their previous drinking habits and its potential impact on the risk of HSV-2 infection. Secondly, despite utilizing regression models and stratified analyses to mitigate the effects of confounding variables, the potential for residual confounding from unmeasured or unknown factors cannot be entirely eliminated. This limitation underscores the need for cautious interpretation of the associations observed, as other unaccounted variables might influence the outcomes. Thirdly, our findings, based on a survey of U.S. adults, may not be generalizable to other populations without further investigation. Cultural, social, and behavioral differences in various populations might affect the relationship between alcohol consumption and HSV-2, thus limiting the universality of our results. Lastly, the inherent limitations of cross-sectional studies constrain our ability to establish a causal relationship between alcohol consumption and HSV-2 infection. Longitudinal studies are required to confirm these findings and to explore the temporal sequence between alcohol consumption and HSV-2 infection. Moreover, although our study focuses on the association between alcohol consumption and HSV-2, other lifestyle factors such as smoking also play a significant role in HSV-2 infection and warrant further exploration to fully understand the multifactorial nature of this health issue.

3.Additional Subgroup Analysis: Consider including additional subgroup analyses based on other potential confounders such as socioeconomic status or geographic location to further explore the heterogeneity of the results.

Response:

Thank you for your suggestion. We have already included 'family income' as a variable in our existing subgroup analyses, as shown in Figure 2. Based on the variables available in our baseline data, we have incorporated additional potential confounders into our subgroup analyses, including race, educational level, BMI, insurance status, smoking status, frequency of condom use, and conditions such as CKD, COPD, CVD, and cancer. There were no interaction effects within these subgroups; the association between alcohol consumption and HSV-2 infection remained consistent across all. The results of these new subgroup analyses have been uploaded as a supplementary file named "Supplementary Figure 1."

Supplementary Figure 1 The relationship between alcohol consumption and HSV-2 in various subgroups

4.Strengthening the Discussion: The discussion section could be strengthened by incorporating a more detailed comparison with existing literature, especially regarding the biological mechanisms by which alcohol consumption might increase the risk of HSV-2 infection.

Response:

Thank you for your suggestion. We have further enhanced the discussion section by conducting a more detailed comparison with existing literature on the topic and introducing new references (citation 22). Specifically, we have expanded our discussion to include an in-depth analysis of the biological mechanisms through which alcohol consumption may increase the risk of HSV-2 infection. This includes an examination of how alcohol impairs immune function, influences behaviors that increase infection risk, and potentially exacerbates the viral replication process. We believe that these additions will provide a comprehensive understanding of the interaction between alcohol consumption and HSV-2 infection, aligning our findings with established research in the field.

Line 244-253：Moreover, studies have shown that chronic alcohol consumption can alter the transcriptome of bone marrow-derived monocytes, potentially impacting monocyte activity and their migration from the bone marrow, thereby affecting immune responses. Additionally, chronic alcohol intake can bias the differentiation of CD34+ progenitor cells in the bone marrow towards the granulocyte/monocyte lineage, leading to a decreased antigen-presenting capability of monocytes derived from CD34+ progenitor cells[22]. This alteration persists even after cessation of alcohol consumption and may compromise both immune and inflammatory responses. These impacts on the immune system and inflammatory responses could increase susceptibility to HSV-2, also explaining why former drinkers may still exhibit a heightened risk of HSV-2 infection.

5.Consistency in Data Presentation: Ensure that tables and figures are consistent with the text and that they are clearly and accurately referenced within the manuscript.

Response:

Thank you for your suggestion. We have carefully reviewed the description of table data and results section, removed redundant content, ensuring that the data in the tables can be clearly and accurately referenced within the manuscript.

Reference Updates: Check the references for the most current and relevant literature to support the study's findings and arguments.

Response:

Thank you for your suggestion. We have carefully checked for the most recent and relevant literature and have added a reference published in 2023 (citation 22) to further support our study's findings and arguments.

6.Language and Grammar: While the manuscript is generally well-written, there may be minor grammatical errors or awkward phrasings that could be polished for better readability.

Response:

Thank you for your constructive feedback on the language quality of our manuscript. We fully acknowledge that being non-native English speakers does not justify the language issues. In response to your feedback, we have made every effort to improve the clarity and readability of our text. We have received assistance from a native English speaker from the United States to thoroughly improve our article. We hope that these revisions meet your expectations.

7.Confirmation of Ethical Compliance: If applicable, provide a statement confirming that the study complies with ethical standards regarding the protection of personal data and privacy of participants.

Response:

Thank you for your suggestion. We have provided a statement confirming that the study complies with ethical standards regarding the protection of personal data and privacy of participants. The confirmation document is attached and is named "Ethical Statement."

---

## [Editor Report · Decision Letter 1]

10 Jul 2024

The association between alcohol consumption and Herpes simplex virus type 2: A cross-sectional study from National Health and Nutrition Examination Survey 2009-2016

PONE-D-24-11862R1

Dear Dr. Shan,

We’re pleased to inform you that your manuscript has been judged scientifically suitable for publication and will be formally accepted for publication once it meets all outstanding technical requirements.

Kind regards,

Tinashe Mudzviti, MPhil(MD)

Academic Editor

PLOS ONE
---

## [Editor Report · Acceptance letter]

15 Jul 2024

PONE-D-24-11862R1 

PLOS ONE

Dear Dr. Shan, 

I'm pleased to inform you that your manuscript has been deemed suitable for publication in PLOS ONE. Congratulations! Your manuscript is now being handed over to our production team.

Kind regards, 

on behalf of

Dr. Tinashe Mudzviti 

Academic Editor

PLOS ONE